# Potential Mechanisms of Action and Outcomes of Equine-Assisted Services for Veterans with a History of Trauma: A Narrative Review of the Literature

**DOI:** 10.3390/ijerph20146377

**Published:** 2023-07-16

**Authors:** William R. Marchand

**Affiliations:** 1VISN 19 Whole Health Flagship Site, VA Salt Lake City Health Care System, 500 Foothill, Salt Lake City, UT 84148, USA; william.marchand@va.gov; Tel.: +1-801-582-1565 (ext. 1847); 2Department of Psychiatry, School of Medicine, University of Utah, 501 Chipeta Way, Salt Lake City, UT 84108, USA; 3Animal, Dairy and Veterinary Sciences, Utah State University, 4815 Old Main Hill, Logan, UT 84322, USA

**Keywords:** veterans, psychiatric disorders, equine-assisted services, PTSD, psychotherapy incorporating horses, military sexual trauma

## Abstract

Equine-assisted services (EASs) are being increasingly used as complementary interventions for military veterans who have experienced trauma. However, there is limited evidence of benefit for this population and almost no literature describing the desired potential outcomes and possible mechanisms of action. The aim of this article is to address these gaps by reviewing the extant literature of animal-assisted interventions in general, and equine-assisted services in particular, with the goal of providing guidance for future investigations in the field. Currently, the field is in the early stage of scientific development, but published results are promising. Interventions that enhance treatment compliance and/or outcomes could benefit this population. Preliminary results, reviewed herein, indicate that EAS interventions might benefit the military veteran population by enhancing treatment engagement and therapeutic alliance, as well as by contributing to symptom reduction and resulting in various transdiagnostic benefits. It is recommended that future studies include exploration of potential beneficial outcomes discussed herein, as well as investigate suggested mechanisms of action.

## 1. Introduction

The aim of this article is to review the extant literature regarding mechanisms of action and associated outcomes of using equine-assisted services for veterans who are trauma survivors. Based on the literature, recommendations are made to advance the scientific development of the field.

### 1.1. Consequences of Trauma Exposure

Trauma exposure occurs commonly worldwide, including from war, disasters, pandemics, and interpersonal assaults [1]. Subsequently, post-traumatic stress disorder (PTSD) is thought to impact approximately 3.6% percent of the world’s population [2] and thus is a significant public health concern. The lifetime prevalence of PTSD among adults in the US general population is estimated at 6.8% [2,3]. However, among military personnel and veterans, rates of PTSD approach 30% [4,5].

In addition to the symptoms of PTSD, this condition is associated with impairments in social, occupational, and physical functioning, as well as reduced quality of life and physical health problems [3]. Functional impairment is exhibited across social, interpersonal, physical health, and occupational domains. This often manifests as poor social and family relationships, absenteeism from work, lower income, and lower educational and occupational success. Individuals with PTSD frequently experience at least one other comorbid mental disorder and are at risk of suicidal thoughts, suicide attempts, and death from suicide [3]. Among veterans with PTSD, up to 80% may have complex PTSD [6], which further increases the risk of suicide and psychiatric comorbidities [7,8]. Lastly, individuals with pain comorbidity have worse outcomes than those with chronic pain alone [9].

The core feature of PTSD is an amplified fear response, which can be thought of as an inability to distinguish between environmental cues that are dangerous versus those that are safe [10]. Studies suggest this is due to abnormalities of both fear learning and extinction [10]. Neuroimaging studies have attempted to define the underlying neural mechanisms with limited success [10]; however, a meta-analysis [11] of functional magnetic resonance imaging (fMRI) studies reported increased fear circuit activation, including the amygdala, during all phases of fear learning and extinction. MRI resting-state functional connectivity studies of PTSD have also yielded variable results [10] but suggest that PTSD is associated with abnormalities of the default mode, salience, and executive control networks [12,13].

### 1.2. Conventional Interventions for Trauma Exposure

Both psychological and pharmacological treatments are available for PTSD. Effective evidence-based psychotherapy interventions for PTSD exist and fall into two broad categories, which are past-focused and present-focused evidence-based treatments [14]. Past-focused models facilitate exploration of trauma in detail to promote the processing of distressing emotions, painful memories, and beliefs about the trauma. Examples include cognitive processing therapy (CPT), prolonged exposure (PE), eye movement desensitization and reprocessing (EMDR), and narrative exposure therapy. Present-focused models emphasize psychoeducation and coping skills to improve current functioning in domains such as interpersonal, cognitive, and behavioral skills. Cognitive therapy for PTSD, seeking safety, and stress inoculation training are examples of present-focused approaches. Research indicates that these interventions work better than treatment as usual with effect sizes in the moderate to high range [14] and that both past- and present-focused approaches appear to work equally well [14]. However, given many studies of past-focused models, these interventions, such as PE, CPT, and EMDR, are often referred to as gold-standard therapies [15]. Although effective, one-third to one-half of veterans receiving exposure-based treatments for PTSD demonstrate no clinically significant improvement [16]. Another important issue is their level of utilization, as well as retention versus dropout. For example, one study found that among post-9/11 veterans, only 23–40% of those screening positive for a probable mental health disorder had sought care [4]. Further, several Veterans Healthcare Administration (VHA) studies have reported high attrition rates for both PE and CPT [17,18]. For example, in a study [17] of 1924 VHA patients who attended at least one session of CPT or PE, the median number of sessions attended was five, with an “adequate dose” defined as eight sessions or more. Another study [19] found that among Iraq and Afghanistan War veterans who had PTSD, only 23% initiated evidence-based psychotherapy and of those who did, only 9% completed treatment.

A variety of factors have been proposed to contribute to treatment resistance among veterans. These include stigma, fear of confronting trauma experiences, the number and frequency of sessions involved in these interventions, concerns about confidentiality, compromised relationships with therapists, and fear of being seen as weak [20,21,22].

Finally, regarding pharmacotherapy, only two antidepressants, sertraline, and paroxetine, are approved for PTSD treatment by the US Food and Drug Administration [23]. While these medications may be beneficial, some patients do not improve, experience bothersome side effects, and/or discontinue treatment [24]. For example, one study [25] of pharmacology for PTSD reported that 35% discontinued treatment within 30 days and 72% discontinued within 180 days.

In addition to challenges related to incomplete treatment response, treatment seeking, and engagement, several other factors may limit the benefits of conventional psychotherapy interventions for some veteran trauma survivors. The first is that rates of military sexual trauma (MST) can be as high as 15% among female veterans [18], and conventional interventions may not address hallmark features and complex aftereffects of MST, which include, but are not limited to PTSD [16]. Additionally, conventional interventions may not fully address the psychological impacts of experiencing the transgressive acts of war. One impact is trauma-related guilt and associated beliefs about the trauma [26]. Considerable evidence demonstrates a link between beliefs about traumatic incidents and the ability to recover [27,28]. Further, guilt is associated with cognitive distortions and disengagement and may be an intermediating factor between trauma, depression, and aggressive behaviors among veterans [29]. A closely related concept is moral injury, which is defined as experiencing distress and impairment related to experiences that violate one’s moral beliefs or causes one to question the morality of the world [30]. There is evidence that potentially morally injurious events include failure to act in accordance with one’s personal values, ethical dilemmas, incidents involving injury or harm to civilians, perception of leadership betrayals, friendly fire incidents, and inability to prevent death or suffering [30]. Studies suggest that moral injury is associated with increased risk of both suicide [30,31] and substance use [32]. Finally, disruptions of attachment may be associated with having experienced trauma [33]. For example, one study [34] revealed that male veterans often have an insecure attachment style and exhibit avoidant behaviors. Other studies indicate that PTSD symptom severity is related to insecure attachment [35]. Thus, there is a need for interventions that specifically focus on MST, trauma-related guild, moral injury, and healthy attachment.

Given the relatively high non-response and attrition rates [36,37,38] to conventional interventions, the treatment barrier of stigmatization [39,40] and the psychological impact of MST, disrupted attachment, and transgressions of war, there is a need for interventions for trauma survivors that may enhance treatment engagement and/or outcomes to conventional interventions [41,42]. Further, there is a need to develop and evaluate novel interventions that might be equally or more effective than existing treatments, particularly in the areas of recovery from MST, trauma-related guilt, healthy attachment, and moral injury. Lastly, it has been hypothesized that providing treatment in alternative settings might enhance engagement [26].

In recognition of all the above, animal-assisted interventions (AAIs) are being frequently used as complementary interventions for trauma survivors in general [43,44,45], specifically among veterans [26,33,46,47,48,49,50,51,52,53,54,55,56,57,58,59,60,61,62,63,64]. AAIs might benefit trauma survivors by facilitating enhanced engagement with conventional interventions and/or, after rigorous studies demonstrating benefit, serve as stand-alone evidence-based treatments.

### 1.3. Animal-Assisted Interventions and Equine-Assisted Services

Animal-assisted interventions (AAIs) are a category of intentions aimed at helping humans and which utilize various animals. Equine-assisted services (EAS) is an umbrella term for a group of horse-related AAIs aimed at providing benefits for human participants [65]. EAS interventions include psychotherapy incorporating horses (PIH), equine-assisted learning, and therapeutic riding [65]. Equine-assisted services are being increasingly used as complementary interventions for both civilian [66] and military [47,49,60,61,64] trauma survivors. However, the field is in the early stages of scientific development and rigorous research is lacking [67]. For example, a research report [68] sponsored by the Department of Defense concluded that insufficient evidence existed to determine the effectiveness and safety of EASs for individuals with PTSD, other psychiatric disorders, or suicide risk.

Despite limited scientific evidence of benefit, EAS interventions are being increasingly used for community populations [69], as well as for military service members and veterans with trauma histories [46]. For example, the Equine-Assisted Growth and Learning Association (EAGALA) now has “Military Services Designation” for practitioners of that model of EAS, and in 2019, the VHA was mandated to set aside funds for EASs from its Adaptive Sports Grant program. Lastly, the Professional Association of Therapeutic Horsemanship International (PATH Intl.), accredited centers providing services to veterans, grew from 178 to 267 centers from 2011 to 2020 [70].

Given the state of the field, studies are needed to convincingly demonstrate benefit and disambiguate potential outcomes and underlying mechanisms of action of EAS interventions for veterans with trauma histories. Regarding mechanisms of action, testable hypotheses are needed to facilitate future studies. This article reviews the extant literature regarding outcomes and mechanisms of action and recommends future research directions to move the field forward.

## 2. Literature Review Search Strategy

This narrative review targeted the existing literature on the mechanisms of action and associated outcomes of equine-assisted services for veterans with trauma histories. However, given the limited literature available specific to this topic, articles from the fields of AAIs in general, human–animal bonding, equitation science, and EAS interventions utilized for non-veteran populations were reviewed. A systematic review was not conducted because the field is in the early stages of scientific development and the extant literature does not support a more rigorous review.

Because this was a narrative review without a strict protocol to be followed, a non-systematic electronic search of the literature was conducted primarily in health science databases, such as PubMed. Google Scholar was also used to find a broader number of related articles using plain-language search strategies. Non-standardized inclusion and exclusion criteria were used, again given the fact that the field is in the early stages of scientific development. Regarding the inclusion criteria, studies and reviews were included if they were relevant to the topic. Articles were included irrespective of their designs, and all countries of origin were eligible for inclusion. Lastly, articles or books found outside our search strategy were included if their content was germane to this review. Selected articles were reviewed, and data were extracted and are reported herein. There were no exclusion criteria.

## 3. Key Findings from the Literature

### 3.1. Equine Characteristics

To begin to disambiguate potential EAS outcomes and underlying mechanisms of action among veteran trauma survivors, it is first necessary to review relevant equine characteristics, including those that contribute to the dimensions of horse–human relationships. It is likely that some of the benefits of certain EAS interventions are secondary to the development of a horse–human relationship and possible bonding. Thus, it is important to understand equine characteristics that support the ability to form relationships and emotional bonding.

#### 3.1.1. Prey Animals

One of the key drivers of equine behaviors is that unlike humans and common domesticated species, such as dogs and cats, horses are prey animals [71]; thus, they have evolved to be extremely sensitive to their environment to avoid being eaten. Horses must constantly scan the environment for predators. To manage that requirement, the equine brain has evolved to connect perception directly to action, for example, to whirl and bolt when potential danger occurs [71]. To constantly scan the environment for danger, they have very sensitive hearing, smell, and touch senses [71]. Further, their visual system is different than humans, with eyes on the side of the head; they have a double sided, 340-degree view of the world, with blind spots directly in front and behind [71]. Thus, horses literally see the world differently and less clearly than humans, and as prey animals, they perceive the world as a dangerous place. Regarding EASs, the fact that equines are prey animals is an important consideration for safety given horses’ predisposition to move quickly when frightened. Further, their sensitivity to the environment means they are very sensitive to humans and may notice the smallest details of human body and verbal language, as well as sense a human’s emotional state. In fact, it has been suggested [72] that horses may have developed some level of emotional intelligence.

#### 3.1.2. Herd Animals

Another key driver of equine behavior, and one very important for EASs, is that horses are herd animals [71]. Thus, they have evolved to depend on relationships with other conspecifics for survival. Horses are very social animals and rely on group perception, learn by imitation, seek leadership from dominant guides, and soothe themselves through social contact [71]. Natural selection has resulted in equine behavioral strategies that promote social stability and affiliative interactions. It is thought that emotional transmission between individuals contributes to group coordination and bonds between individuals [72]. For example, transmission of positive valenced emotions could contribute to group synchronization, whereas the rapid transfer of negative ones, such as fear, may enhance survival. Given their social nature, horses are very capable of developing long-term relationships with humans.

#### 3.1.3. Horse Cognition and Emotion

Equine cognition has likely evolved to facilitate both prey and herd animal behaviors. The human brain represents 2% of total body weight and uses 20% of the body’s glucose [71]; in contrast, equine brains comprise only two-tenths of a percent of a horse’s total body weight but use 25% of the glucose [71]. Further, the horse brain contains far fewer neuros (1 billion neurons compared to 86 billion for humans) [71]. Nonetheless, equines have excellent memories and extensive learning abilities [71].

Equine cognition facilitates herd behavior, including their ability to discriminate relatives from non-relatives and recognize specific individuals [72]. Similarly, horses can recognize individual humans [73,74] and are thought to form expectations of individuals based on past personal experience with that individual [72]. Further, the reactions of horses towards people in general are the result of factors including specific experiences with humans, the equine’s personality and temperament, as well as the temperament and skills of the humans with which it interacts [75]. Horses can respond to many human communication cues, for example, horses can respond to human pointing, which requires cognitive skills that facilitate referential communication [72]. Finally, it has been proposed [72] that horses may have developed some level of emotional competence, like human emotional intelligence, which could facilitate recognizing and thus appropriately reacting to a human’s emotional state. At least one study [76] provided evidence for this by demonstrating that horses can form long-term memories of specific humans based only upon previous observation of these individuals’ emotional expressions in pictures. Thus, horses appear to be able to develop relationships with humans based upon both memory and emotion.

Mammal emotion can be understood by the field of affective neuroscience [77]. Research has identified seven primal emotional states, which appear to exist to promote survival [77]. These emotions arise from subcortical brain regions that are largely homologous among mammals [77]; therefore, it is likely that horses and humans experience basic emotions similarly [72]. The positive valenced primal emotions are seeking, lust, care, and play, while those with a negative valence are fear, sadness, and anger [77]. Horses communicate their emotional state using body language, including facial expression. This includes a repertoire of facial expressions, comparable to other animals, such as primates and dogs [78]. Lastly, horses may be able to convey emotional states using vocalizations and to perceive variation in vocal parameters accounting for emotional valence [79]. In summary, it is likely that horses and humans share an emotional language with some commonalities and may be able to accurately read each other’s emotional states. This ability may be a primary foundation underlying the benefits associated with EAS interventions.

### 3.2. Potential Mechanisms of Action and Outcomes of EASs for Veterans with Trauma Histories

It is likely that various mechanisms of action underlying the benefits of EASs operate simultaneously and synergistically. Also, it is likely that different EAS interventions may have different mechanisms of change and outcomes. Furthermore, as we have previously reviewed [67], a major challenge in the field is the lack of standardization of both interventions and terminology, making interpretation of the existing literature challenging. Nonetheless, to advance the science of the field, it is necessary to investigate potential outcomes and mechanisms separately, and then try to disambiguate synergistic effects. To that end, this review aims to summarize what is currently known about mechanisms of action and outcomes and suggest relevant research strategies to move the field forward.

#### 3.2.1. Overview of the Extant Literature

Many EAS studies are from studies of non-veteran samples. Psychological benefits have been reported for a variety of disorders, including autism spectrum disorders [80,81,82,83,84,85,86,87,88,89], schizophrenia spectrum illness [90,91,92,93], social anxiety [94], attention-deficit/hyperactivity disorder (ADHD) [95,96,97], dyspraxia [98], attachment disorders [99], and depression [100]. Studies have also reported improvements in quality of life, cognition, and well-being [69,101,102,103].

Regarding non-veteran trauma survivors, EASs have been associated with reduced symptoms of depression, anxiety, externalizing behaviors, and PTSD symptoms among children and adolescents who have experienced trauma [66,104,105,106]. Lastly, a meta-analysis of EASs for at-risk adolescents with trauma histories found a medium effect size for seven investigations [107]. Taken together, these investigations suggest that EAS interventions may provide psychological benefits for some children and youth with trauma histories. However, rigorous research is generally lacking, and further studies are needed [67]; furthermore, it is unknown if these results generalize to veteran populations.

There is now emerging literature reporting investigations of EAS interventions for veterans with trauma histories. This review identified 23 studies [26,33,46,47,48,49,50,51,52,53,54,55,56,57,58,59,60,61,62,63,64,108,109,110,111] in the literature (Table 1). However, of these, only three [47,52,53] had a control group and only one [52] was a randomized trial. There is also a case study [112] of a single veteran who experienced improved psychological functioning associated with participating in horsemanship training (not included in Table 1). Most studies report on US military veterans and their sample sizes range from five to eighty-nine veterans (some studies include couples with non-veteran partners). The majority report quantitative data, but five [33,49,54,108,109] report qualitative data or mixed methods. Four studies [47,55,63,110] report physiologic outcome measures (discussed in more detail below). Lastly, several studies [26,46,53,57,58,63] report interventions that are manualized or structured to facilitate manual development. All the studies suggest benefit for veterans, as is reviewed in a subsequent section of this paper; however, the lack of rigorous research indicates that the field is still in the early stage of scientific development. The aim of this review is to outline potential outcomes and mechanisms of action to help move the field forward.

#### 3.2.2. Horse–Human Relationships, Attachment, and Bonding

An important feature of some EAS programs is the opportunity for the participant to form a relationship with an equine [113]. Human–horse relationships are thought to contribute to benefits associated with EASs in general [114], and several investigators [54,111,112] have theorized that human–horse bonding contributed to outcomes in studies of veterans. Further, it has been hypothesized that the horse could serve as an attachment figure in some therapeutic situations [115,116]. Attachment is often defined as a social connection between two living beings that is characterized as providing feelings of safety [117]. Persistence of attachment may lead to bonding, which implies an ongoing close and interactive relationship between individuals [117,118]. It is thought that attachment and bonding evolved as a means of keeping offspring close to their caregivers for the survival of the offspring [117]. One definition of a human–animal bond is that it is a dynamic and mutually beneficial relationship between a human and a non-human animal, which is modulated by reciprocal interactions essential to the health and well-being of both [119]. These interactions can have emotional, psychological, and physical components and thus could contribute to healing for the participant, the equine, or both [72]. Thus, it is important to understand how horse–human relationships might result in attachment and bonding, as well as the implications for EAS interventions.

Horses have shared about 5500 years of co-evolution with humans, thus allowing the growth and establishment of horse–human interspecific relationships [72]. The domestication of horses likely resulted in equines developing skills to relate to, and interact with, humans [120]. These skills appear to include a high level of sensitivity to human cues and emotional intelligence [72], as well as the ability to form higher-order concepts of specific humans [121]. Thus, horses and humans can clearly form relationships, and the benefits of some EAS interventions are thought to be related to emotional connectedness and effective horse–human relationships [72].

Effective human–animal relationships are characterized by the exchange of reciprocal behaviors across repeated encounters [72]. This communication through a cross-species platform and the language used in horse–human dyads is thought to be primarily non-verbal, conveyed via both physical touch and emotional connection [72]. Horse–human physical contact may serve as a channel for both coordination of motor activities and emotional connection [72] and likely plays a key role in bond formation [118]. Horses naturally groom each other, and this behavior has been shown to be associated with decreased heart rate [122]. Similarly, human touch can lead to a heart-rate reduction in equines [123], which suggests that human physical contact can be calming for equines. Thus, it is likely that touching and proximity play key roles in relationship development and bonding [72]. Further, it has been hypothesized that interspecific emotional transfer occurs between equines and humans during EAS interventions, which may include coupling of emotional states [72].

Further research is needed to fully disambiguate horse–human attachment and bonding, and it cannot be assumed that horses develop human-like attachment relationships to people [118]. As reviewed by Payne et. al. [118], human attachment is characterized by proximity seeking, sense of a safe-haven, secure base, and separation distress. For the horse–human dyad, there is some evidence for proximity seeking by equines [118], limited evidence of the safe-haven effect [118,124], and no evidence of the secure-base effect. Thus, while horses and humans can clearly form relationships, further research is needed to disambiguate horse–human bonding and attachment on the equine side of the dyad.

The horse–human relationship might contribute to healing and recovery in several ways. First, the experience of trauma can significantly disrupt how individuals attach to and bond with other humans [125]. Horse–human relationships may facilitate a safer form of attachment [126] as trauma survivors may trust a therapy animal more readily than a human [127]. Also, animals may provide direct social support through nonjudgmental support and unconditional positive regard [126]. It is thought that the development of human–animal attachment can facilitate a transition to feeling more security in human relationships [126] and ultimately facilitate human–human attachment. Trauma-related impairment of attachment often includes social isolation and an inhibited ability to engage in physical touch with humans [3]. Traditional psychotherapy does not provide an opportunity for social connection or touch; in fact, it is strictly prohibited. In contrast, EASs may not only facilitate connection and attachment with another living being, but also provide opportunities for physical touch and affection between the participant and equine.

Another posited mechanism of benefit is the enhancement of a sense of control, autonomy, and assertiveness for participants [111,112] that occurs because of the horse–human interaction. Many EAS interventions include horse handling and/or riding. These activities require the human to take on a leadership role [69]. This can be challenging because the size of a horse can be intimidating, and many individuals find it difficult to be assertive. By taking a leadership role, participants can not only have a sense of control and autonomy, but also practice appropriate assertiveness in a safe environment.

Lastly, some EAS programs may allow an opportunity for animal care, such as brushing and feeding [128]. Participants might feel an enhanced sense of purpose if they take on some responsibility for equine care.

In summary, it is likely that horse–human relationships contribute to benefits associated with EAS participation. However, much additional research is needed to disambiguate the mechanisms of action. Studies focused on characterizing horse–human relationships and associated benefits are warranted to move the field forward.

#### 3.2.3. Treatment Engagement and Therapeutic Alliance

One way that EAS interventions could benefit veterans with trauma history would be to facilitate engagement in conventional psychotherapy and/or psychopharmacological interventions. In addition to the facilitation of engagement in conventional interventions, trauma survivors might be more likely to engage in EAS interventions than conventional treatments. If so, then it is crucial to conduct rigorous studies to determine if these interventions have benefits comparable to standard treatments for trauma.

First, there is the literature in the larger field of AAIs that informs this topic. There is consensus in the AAI field that these interventions enhance rapport building between client and therapist [126,129] and support the formation of a therapeutic alliance [128]. These effects could be the result of an animal’s ability to create a relationship with the participant(s), offer affection, and provide a less-threatening opportunity for connection with both therapists and the intervention [130]. Also, clients assess therapists for the level of psychological safety of working with the person [113]; thus, observing a therapist work with an animal in a positive and respectful way may enhance trust of the therapeutic relationship [113]. These effects on engagement can be particularly important for trauma survivors for whom lack of trust is common [113,128]. Additionally, given that animals may be perceived as being more non-judgmental than humans [128]. As stated above, trauma survivors may trust a therapy animal more readily than a human, which can serve as an antecedent to developing trust with a human therapist [127]. Further, animals may motivate participations by way of activation of implicit motives via the experiential human–animal interaction and thus promote intrinsic motivation [128], which promotes task enjoyment and satisfaction. The casual setting of an equine facility may be more comfortable for clients with trauma histories than a therapist’s office [113]. The potential enjoyment of the EAS experience, along with the casual, naturalistic setting likely contribute to treatment engagement. Lastly, AAI-induced activation of the oxytocin system (discussed below) may enhance the development of trust toward the therapist early in the therapy, decrease anxiety, and improve motivation [128].

Regarding studies of EASs for veterans, there is evidence that participants find EAS interventions to be enjoyable [57,58], which may enhance treatment engagement. One study [58] found that among participants in a six-session intervention, 58% completed four or more sessions and 24.2% completed all sessions. Another study [57] found that for a four-session intervention, the mean number of sessions attended was three and that 52% of participants completed all sessions. These results compare favorably with studies of veteran utilization of conventional psychotherapy. For example, one study [131] found that only 34% for group psychotherapy and 12% for individual psychotherapy received an adequate dose. Another investigation [132] revealed that among women veterans with PTSD, only 42% received a minimal therapeutic dose.

These studies suggest promise; however, no studies have directly addressed whether EAS interventions may enhance engagement and/or utilization of conventional interventions for PTSD. Future studies will need to focus specifically on whether, and how, EASs may facilitate treatment utilization and engagement among veterans with trauma histories.

#### 3.2.4. Transdiagnostic Benefits and Symptom Reduction

Evidence suggests that AAIs in general, and EAS interventions specifically, are associated with both transdiagnostic benefits, as well as symptom reduction. One such transdiagnostic benefit is decreased arousal. For example, there is compelling evidence in the AAI literature that the presence of an animal can result in decreases in physiological indicators of arousal, such as skin conductance, blood pressure, and heart rate among non-traumatized human populations [133,134,135,136]. Canine–human interactions have been particularly well studied in this area. For example, a systematic review [137] reported a significant reduction in heart rate, blood pressure, and/or cholesterol levels in some studies. Individual studies have reported decreased heart [138,139,140,141] and respiratory [138] rates, as well as lowered blood pressure [140] and increased heart-rate variability [139]. AAI-associated neuroendocrine changes (discussed below) are thought to, at least partly, underlie decreased stress and arousal associated with these interventions [126]. Other transdiagnostic benefits that have been reported in non-veteran populations include improvements in cognition, quality of life, and well-being [69,101,102,103]. Regarding arousal, one study of veterans reported decreased blood pressure [55]. Studies of veterans suggest transdiagnostic benefits of decreased disability [53] and improved functioning [54,64]. Finally, one study [47] reported increased resilience, but another [51] found no change in this measure.

In addition to transdiagnostic benefits, the literature suggests that EASs are associated with a reduction in symptoms associated with specific psychiatric disorders. As stated above, studies of community populations have reported benefits for a variety of psychiatric conditions [80,81,82,83,84,85,86,87,88,89,90,91,92,93,94,95,96,97,99,100,142,143], including trauma-related symptoms [66,104,105,106,107]. Veteran-specific studies have reported an improved effect [51,56,57,58,109,110], and psychological flexibility [57,58], as well as decreased anxiety [51,56,59,60], depression [46,50,53,57,58,59,60,61,63], craving for substances [51,56], and PTSD symptoms [26,46,47,48,50,52,53,54,55,59,60,61,63,64]. Other reported outcomes include that the participants enjoyed the activity [57,62] and that the activities resulted in no adverse effects [46,51,52,56,57,58,64]. Taken together, these studies suggest that EAS participation has a low risk of adverse outcomes, as well as being associated with transdiagnostic benefits and symptom reduction among veteran trauma survivors. However, much more rigorous studies are needed. Only one study in the literature was randomized [52], and while demonstrating benefit, its limitations include a moderately small sample size. Two studies [47,53] had a control group and reported improvements in PTSD symptoms, but one [47] found no differences between the treatment and control groups. Thus, additional research is needed to fully establish the benefits of EASs for veterans.

#### 3.2.5. Emotional Mirroring and Heart-Rate Synchronization

As stated above, horses are prey animals and therefore need to elude predators for survival [71]. Thus, horses have evolved to become exceptionally sensitive to, and aware of, their environment. This includes heightened awareness of other nearby animals and humans [71]. This includes extreme sensitive to inconsistencies, agitation, and autonomic arousal among nearby animals [26], all of which could indicate an imminent attack by a predatory animal. Thus, horses can sense and mirror human emotional states [128]. The sensitivity of horses to human emotions may facilitate the therapeutic process by providing the human participant with feedback regarding their actions, emotion state, and body language, which may enhance human insight and self-awareness. Emotional mirroring may also offer humans a way to examine and discuss their emotions without feeling overwhelmed or concerned about being judged by the therapist. More research is needed in the EAS field, but it is likely that in some circumstances, horses could play a role like some psychiatric service dogs that provide a biofeedback function alerting humans to anxiety or arousal and thus facilitating appropriate self-care [126]. Lastly, there is preliminary evidence [144,145,146,147] that horse–human heart-rate synchronization may occur during some, but not all [145], interactions, which may be related to equine sensitivity to the environment. This phenomenon warrants further study as it may contribute to the therapeutic response experienced by humans.

#### 3.2.6. Self-Distancing through Metaphor

Some PIH interventions, such as the Eagala model (https://www.eagala.org/index, accessed on 4 April 2030), incorporate the use of horses, arena props, and the equine facility setting as a metaphor in an experiential, solution-focused learning process [148,149]. Metaphor facilitates making the participant’s emotions and cognitions tangible through physical and visual representation [113] and allows self-distancing. For example, a participant who observes two horses standing close together might perceive that as a representation of a close relationship in their own life, which is creating challenges. Because of self-distancing through metaphor, a participant may be able to work safely with emotions and cognitions associated with the relationship. Thus, this process can result in psychological insight for the participant and may include the opportunity to experience and practice new behaviors in a low-threat, metaphorical setting [113]. Studies of community [150,151,152] and veteran [47,61] populations are promising, and thus rigorous studies of Eagala model interventions among veterans are warranted.

#### 3.2.7. Psychological Flexibility, Biophilia, and Mindfulness

Some evidence [57,58] suggests that enhanced psychological flexibility may be associated with participation in equine-assisted services. Psychological flexibility can be defined as pursuing goals, despite feeling distress [153]. Another definition [154] of psychological flexibility is that it describes how an individual adapts to shifting situations, shifts perspective, and balances competing desires and needs. Lastly, psychological flexibility has been defined as a mindful orientation, in which thoughts and feelings are recognized as such facilitating value-based actions that are incongruent with short-term cognitions and emotions [155]. Acceptance and commitment therapy is a type of psychotherapy that enhances psychological flexibility [155], and studies of acceptance and commitment therapy indicate that improved psychological flexibility is associated with decreased substance abuse, depression, self-harm, chronic pain, anxiety, prejudice, and stress, as well as other positive outcomes [155]. Further, psychological flexibility is an important dimension of overall psychological health [154] and is closely related to resilience [154,156,157].

In addition to the limited evidence from equine-assisted services studies that have measured psychological flexibility [57,58], enhanced psychological flexibility is a candidate outcome underlying some of the transdiagnostic benefits of equine-assisted services discussed above. This is because the literature indicates that animal-assisted interventions in general positively impact resilience [134,158], as well as other outcomes closely related to psychological flexibility, such as decreased arousal, cortisol response, stress, and burnout, as well as increases in compassion and positive effects [133,159,160,161,162,163]. Equine-assisted services studies have also reported enhanced resilience [47,164,165], as well as other outcomes associated with psychological flexibility, including a reduction in depressive [46] and anxiety [66,94] symptoms, as well as an enhanced quality of life [60,102,103,166,167,168] and interpersonal interactions [169]. Thus, future studies of various equine-assisted service interventions may benefit from assessing changes in psychological flexibility. Lastly, potential neural mechanisms underlying increased psychological flexibility and decreased experiential avoidance have been elucidated with functional magnetic resonance imaging (fMRI) studies [167,168], and an association between 5HTT polymorphism and psychological flexibility has been reported [170], thus implicating serotonin in our understanding of neural mechanisms of psychological flexibility. Therefore, future studies of changes in psychological flexibility associated with equine-assisted service participation should include physiologic mechanisms contributing to positive outcomes.

Relatedly, biophilia may play a role in equine-assisted service outcomes. Studies [171,172] on nature exposure for veterans have revealed enhanced psychological flexibility, and nature exposure is generally associated with a variety of benefits like those associated with increased psychological flexibility, including enhanced resilience [173]. The mechanism of these benefits is often explained via the concept of biophilia [173], which refers to human affinity to nature in general, including animals. Biophilia is thought to have developed during evolutionary history as a mechanism to enhance the survival of the human species [173,174]. The biophilia effect [127,174] describes the phenomenon that some humans experience psychological and physiological relaxation around calm, resting animals [128]. It is thought that at a subconscious level, humans perceive relaxed animals as a signal of a safe environment [128].

Lastly, the concept of mindfulness warrants a brief discussion because there is evidence that exposure to nature can enhance mindfulness [175], and there is a close relationship between psychological flexibility and mindfulness [176]. Further, it has been hypothesized that animal-assisted interventions can be an effective form of mindfulness training [128]. Working with horses provides a compelling reason to focus on the present [113] and provides equine-assisted service participants a real-world opportunity to practice mindful awareness.

In summary, the literature suggests that psychological flexibility, biophilia, and mindfulness are highly related concepts and that all may play a role in the therapeutic benefits of equine-assisted services. Future studies will be needed to fully disambiguate the role and interdependence of these mechanisms in equine-assisted service outcomes. Such studies should consider incorporating functional neuroimaging and/or serotonin assessments.

#### 3.2.8. Physiologic Outcome Measures

In addition to research utilizing psychological instruments, studies utilizing physiological measures could provide insights into the mechanisms of EAS interventions. Physiologic outcomes that have been studied in AAIs and show promise for future studies include measurements of cortisol, oxytocin, heart-rate variability (HRV), and brain activation.

Cortisol is a glucocorticoid hormone produced by the adrenal cortex. Sympathetic nervous system activation stimulates cortisol secretion to enhance energy to protect biochemical processes during stress. Chronic levels of elevated cortisol result in suppression of cellular and humoral immunities [16], and thus chronic cortisol elevation is harmful to the human body. In the AAI literature, there is evidence of attenuated cortisol response associated with interventions involving dogs in some [177,178,179,180,181,182], but not all, studies [138,183]. In a recent systematic review of canine studies, Rathish and colleagues [137] reported that a few studies showed a significant reduction in cortisol. However, a study [183] of military veterans with PTSD revealed those with service dogs had higher cortisol awakening response compared to the waitlist. Regarding studies of EASs, one study [184] of a youth population found a decrease in cortisol at timepoints but no overall pre- to post-test changes. In contrast, one study [47] of an EAS intervention for veterans found no change in cortisol response associated with the intervention. While studies thus far have revealed mixed findings, further research into using cortisol as a physiologic marker for EAS investigations is warranted. Given that cortisol can be effectively collected and measured in studies of equines, for example [55,185], this measure can be used to evaluate both horse and human responses to EAS interventions.

Oxytocin is a hypothalamic neuropeptide composed of nine amino acids [186], which is involved in multiple complex physiological and behavioral functions. As reviewed by Beetz and colleagues [128], activation of the oxytocin system could explain many of the positive benefits of AAIs. Oxytocin has classical hormonal functions as it is stored in the posterior pituitary, and when activated, it is released into the blood stream where it is carried to target tissues throughout the body [186]. However, it also functions as a neurotransmitter [128,186]. Oxytocin is well known for its essential function in birth and lactation. However, it can also be considered a stress-modulating hormone as it impacts stress reactivity, decreases anxiety, and facilitates recovery following periods of difficulty [128,186]. Stress modulation occurs via interfaces with the immune and autonomic systems, as well as the hypothalamic–pituitary–adrenal axis [186].

Oxytocin’s role in mammalian behavioral functions has relevance for AAIs [187]. Oxytocin can inhibit the defensive actions of vasopressin and other stress-related pathways and thus reduce the perception of threat, allowing animals to engage in prosocial interactions and develop selective relationships and bonds [186]. Oxytocin is released because of touch, including breastfeeding, massage, stroking, and general pleasant physical contact [128]. Thus, activation of the oxytocin system likely results in similar effects during AAIs, including the promotion of communication, social interaction, trust, and calmness, as well as a reduction in anxiety, stress, and pain [128,186,187]. Oxytocin is thought to exert anxiolytic and stress-reducing actions via modulatory effects on amygdala circuits via serotonin mediation [188]. At least one canine AAI study [139] demonstrated increases in salivary oxytocin, and a non-veteran EAS study [184] found significantly increased oxytocin at some timepoints, but no overall pre- to post-test changes.

More research is needed, but measurement of changes in oxytocin might provide valuable information about mechanisms underlying benefits of EAS interventions. However, concerns have been raised about the validity of peripheral oxytocin assessments in humans, for example [189]. Tabak and colleagues [190] propose solutions that potentially could be applied to EAS studies. Given the known actions of oxytocin, there seems to be a relatively high likelihood that oxytocin modulation may underlie some benefits of EAS; therefore, further research is warranted.

Heart-rate variability (HRV) is another potential physiologic marker that may be associated with the benefits of EAS. HRV is a non-invasive measure of autonomic nervous system (ANS) regulation of cardiovascular function, which can be used to evaluate acute and chronic stress responses in both horses and humans [191,192]. HRV is the naturally occurring irregularity in heart rate, and this intricate beat-to-beat variation results from the interaction between parasympathetic (vagal) and sympathetic branches activity on the heart [192]. During stress, the sympathetic branch is activated and variation between heartbeats is decreased as the heart starts to pump at a regular rate. In contrast, when the parasympathetic branch engages to counter the stress, the variation between heartbeats is higher and the sympathovagal balance is restored [193]. Thus, reduced stress is associated with increased HRV. Sympathovagal balance in humans is associated with enhanced focus, clarity of thinking, and decision making, as well as decreased anxiety [194].

Importantly, human-based HRV frequencies can be used in horses [192], thus facilitating direct comparisons between horse and human HRV patterns and therefore providing a quantitative measure of interactive ANS responses between the two species [192]. Studies of non-veteran populations have revealed that HRV can be measured in EAS interventions [195,196]. A recent systematic review by Garcia-Gomez and colleagues [197] found several studies that reported increased HRV and enhanced parasympathetic activity associated with EAS participation. This suggests a mechanism by which EASs may lead to improved emotional regulation in humans; however, as pointed out by Garcia-Gomez et al. [197], more research is needed. One small non-veteran study [193] demonstrated that it is feasible to measure HRV during EASs for this population. A larger study [110] of a veteran population demonstrated increased HRV was associated with EAS participation. Thus, measurement of HRV may make significant contributions to the understanding of physiological mechanisms underlying the benefits of EASs. Further, HRV assessments can be utilized to further explore the intriguing potential [144,145,146,147], mentioned above, that horse–human heart-rate synchronization may occur in some human–equine interactions.

Lastly, functional neuroimaging, such as functional magnetic resonance imaging (fMRI) has the potential to enhance our understanding of the neural mechanisms underlying the benefits of EAS interventions. For example, one study [198] investigated the effects of an EASs on participants with attention-deficit/hyperactivity disorder by comparing resting-state functional magnetic resonance imaging signals and their clinical correlates. Results suggested that EAS participation may be associated with connectivity changes in the default mode network and the behavioral inhibition system. Another study by Zhu and colleagues [63] used fMRI and diffusion tensor imaging (DTI) to assess EAS-associated changes among veterans with PTSD who participated in an EAS intervention. Resulted indicated a significant increase in caudate functional connectivity, as well as a reduction in the gray matter density of the thalamus and the caudate. The increase in caudate functional connectivity was positively associated with symptom improvement. The authors hypothesized that EASs may target reward circuitry responsiveness. Taken together, these two studies suggest that functional neuroimaging may be a useful tool to investigate neural mechanisms underlying benefits associated with EASs.

## 4. Discussion

As stated above, studies are needed not only to convincingly demonstrate benefit, but also to disambiguate potential outcomes and underlying mechanisms of action of EAS interventions for veterans with trauma histories. It is necessary to identify potential outcomes and develop strategies for rigorous investigations of benefit, as well to develop testable hypotheses of mechanisms of action to guide future studies. The aim of this section is to recommend key research areas and priorities to move the field forward (summarized in Table 2), based upon the literature reviewed herein.

### 4.1. Gaps in Current Conventional Treatment Approaches

As reviewed above, current conventional treatments for veterans who have experienced trauma are, in some cases insufficient, to result in full recovery. EAS interventions have the potential to address this gap. As complementary interventions, EASs might enhance treatment response to, and or engagement with, conventional interventions. Further, if rigorous studies demonstrate benefits, then EASs might serve as a stand-alone treatment for this population.

### 4.2. Current State of the Literature and Manualized EAS Interventions for Veterans

Twenty-three studies [26,33,46,47,48,49,50,51,52,53,54,55,56,57,58,59,60,61,62,63,64,108,109,110,111] were included in this review (Table 1). Of these, only three [47,52,53] had a control group and only one [52] was a randomized trial. Sample sizes varied widely and only four [47,55,63,110] reported physiologic outcome measures. In addition to the fact that rigorous research is lacking, a major challenge for the field is the fact that many different interventions are currently being utilized (Table 1). Further, interventions may include mounted activities only, groundwork only, or both. Also, both group and individual interventions are reported. Lastly, many interventions are not well described in the literature. Thus, drawing conclusions from the existing literature is difficult, other than to conclude that the field is in the early stage of scientific development.

To move the field forward, it is necessary for investigators to study standardized interventions that are designed to be manualized. Manual-based interventions facilitate fidelity and support both large replication studies and ultimately dissemination to the field. A promising development is the published reports of interventions [26,46,53,57,58,63] designed to be manualized. These include equine-facilitated cognitive processing therapy [26], EAT-PTSD [46,63], Riding Through Recovery [53], horsemanship skills/trail riding [57], and Whispers with Horses [58]. Interventions other than the above may been intended for manualization, but this was not stated clearly in the manuscript. EAS investigators should consider studying one of these interventions designed to be manualized to help move the field to a higher stage of scientific development. This strategy would both support the modification of existing interventions based upon research findings and as inform the development of novel interventions.

### 4.3. Dose–Response Relationships and Frequency of Treatment

Currently the dose required to achieve any benefit, or maximum benefit, from EAS interventions is unknown and likely varies with the intervention and participant. Similarly, the ideal frequency of dosing is also unknown. Studies in the current literature regarding EASs for veterans range from one-time interventions [51,56] to four [57], six [58], eight [46,50,53,54,59,63,109,110], or twelve-session [26] interventions, as well as those lasting six weeks [47,52] and six months [64,108] and those offered in retreat style [48,49,55,61,62]. While eight-week and retreat-style interventions are the most common, it is unclear why these lengths of treatment were chosen. Future studies need to compare one intervention across multiple dosing and frequency options to determine the most cost-effective delivery method.

### 4.4. Horse–Human Interaction

Even though EAS interventions are based upon horse–human interactions, the mechanisms of action underlying how these relationships may benefit humans are not well understood. Further, the contribution to beneficial outcomes of mental health treatment incorporated in some EAS models versus that of horse–human interaction is unknown. Also, it is unclear to what extent other factors, such as biophilia related to the equine facility environment, may play a role in contributing to the benefits of EASs. Lastly, the contributions to the benefits of mounted versus groundwork is undetermined.

Hypothesized mechanisms by which horse–human interactions could contribute to benefits for humans include physical touch, interspecific bonding [54,111,112], emotional transfer [72], and decreased arousal, all of which may be related to emotional mirroring, psychological flexibility, and heart-rate synchronization [144,145,146,147]. Additionally, emotional mirroring likely provides a way for humans to examine and discuss their emotions without feeling overwhelmed or judged. Positive outcomes could also be related to the horse serving as an attachment figure [115,116], facilitating self-distancing through metaphor [148,149], and/or providing support through nonjudgmental support and unconditional positive regard [126]. Lastly, enhancement of a participant’s psychological flexibility [57,58], as well as their sense of control, autonomy, and assertiveness [111,112], could be a result of some horse–human interactions.

Regarding benefits attributable to mental health treatment versus horse–human interactions, studies of veteran populations report benefits from EAS interventions that have a mental-health-treatment component, for example [26,46,47,48,49,50,53,56,58,59,60,61,63,108], as well as from interventions that do not, for example [51,52,57]. These latter studies support the hypothesis that there may be a therapeutic benefit for mental health from horse–human interactions even without a specific mental-health-treatment component.

Studies of veteran populations indicate benefits of interventions that include mounted work, for example [52,53,55,57,59,62,64,108,110], and those that are only groundwork, for example [46,47,50,56,58,61], and at least one [51] included only mounted activities. Direct comparisons of standardized interventions with and without mounted work are needed to see if benefits of mounted work justify the additional associated costs and safety risks.

Given the uncertainty of how horse–human interactions result in positive outcomes for humans, rigorous studies of mechanisms of action studies are warranted. These should include measures of participant attachment, psychological flexibility, sense of control, autonomy, and assertiveness. Further, human–animal bonding should be measured. Also, future studies need to directly compare interventions with and without a mental health component, as well as those with and without mounted activities. For interventions with a mental health component, comparisons of those with and without self-distancing through metaphor are needed. Lastly, physiologic measures are needed to fully disambiguate the mechanisms underlying the benefits of horse–human interactions. Potential measures include both horse and human HRV, cortisol, and oxytocin, as well as human blood pressure, skin conductance, and brain activation.

### 4.5. Treatment Engagement and Therapeutic Alliance

Most of the evidence in this area is from the general AAI literature. These studies suggest that AAIs enhance trust [127] and rapport building between client and therapist [126,129], as well as support the formation of a therapeutic alliance [128]. Veteran-specific studies suggest that participants find EASs to be enjoyable [57,58] and that attrition rates compare favorably with studies on veteran utilization of conventional psychotherapy [57,58]. These results are promising, but more research is needed, specifically direct comparisons of enrollment and attrition in EASs versus conventional treatments, as well as the utilization of treatment engagement and therapeutic alliance scales in future studies. Other important topics include determining if physiologic or psychological measures and/or horse–human bonding correlates with treatment engagement and/or therapeutic alliance. Lastly, it is important to determine whether participation in complementary EASs results in improved engagement and therapeutic alliance for conventional interventions for trauma survivors.

### 4.6. Transdiagnostic Benefits and Symptom Reduction

Several potential transdiagnostic benefits have been reported in community samples, including improvements in quality of life, cognition, and well-being [69,101,102,103]. One study of veterans reported decreased blood pressure [55], suggesting decreased arousal. Other studies of veterans suggest decreased disability [53] and improved functioning [54,64]. Finally, one study [47] reported increased resilience, but another [51] found no change.

While for the most part rigorous research is lacking, there is considerable preliminary evidence that EAS participation is associated with decreased psychiatric symptoms in community [90,91,92,93,94,95,96,97,98,99,104,105] and veteran samples. Studies of veterans have reported improved effects [51,56,57,58,109,110] and psychological flexibility [57,58], as well as decreased anxiety [51,56,59,60], depression [46,50,53,57,58,59,60,61,63], and craving for substances [51,56]. Importantly, a reduction in PTSD symptoms has been reported in both community [66,106,107] and veteran [26,46,47,48,50,52,53,54,55,59,60,61,63,64] studies. While these studies are promising, randomized controlled trials are needed to fully establish short and long-term benefits. To establish mechanisms of action, it is important to determine if physiologic or horse–human bonding measures correlate with outcomes.

### 4.7. Potential Adverse Effects of EASs for Humans

There has been minimal discussion of potential adverse effects from EASs for humans in the literature. Horse-related injuries are possible and mounted activities are a concern as they are more dangerous than many other risky activities [199], and at least one case of serious injury has been reported [200]. However, several veteran studies have reported no adverse outcomes [46,51,52,56,57,58,64].

Besides horse-related injuries, re-traumatization is always a risk in therapy or other activities for trauma survivors [113]. This risk must be taken into consideration in EAS session planning [113], including developing a response plan. Lastly, approaching a large mammal, such as a horse, might cause apprehension [113], and some common tasks, such as picking up equine feet or riding, might result in anxiety [113], which could manifest as short-term discomfort or result in re-traumatization. Thus, it is critical that EAS participants be monitored for adverse events and that an appropriate response is initiated if these occur. Additionally, future studies should evaluate for, and report on, adverse outcomes.

### 4.8. Potential Adverse Effects of EASs for Equine Partners

Finally, equine welfare in trauma treatment is a very important consideration [113]. However, little is known regarding the impact of EASs on equines [201]. As discussed above, emotional mirroring [202] and heart-rate synchronization is thought to occur during some EAS activities; therefore, human psychophysiology may have short- or longer-term effects on equine health [202]. Several equine physiologic studies [55,202,203,204,205] suggest that EAS activities may not unduly burden horses with stress; however, much more research is needed. One approach is to evaluate HRV. As a study [193], suggests HRV can be used as a measure of EAS-induced stress in equines. Other potentially useful measures are equine cortisol and behavior scales [55,203]. Since these are useful measures of both horse and human responses to EAS, these modalities could be particularly important for future studies to evaluate both equine stress and mechanisms of action.

Lastly, beyond concern for equine welfare, when horses are stressed, behavior responses occur, including changes in gait, head height, distance from humans or other horses, and ear orientation. This could result in injury to participants and/or staff. Awareness and understanding of equine responses to stress facilitates safety, for both horses and humans, while working together in therapeutic activities.

### 4.9. Limitations of This Review

There are several limitations that must be considered when interpreting this review. First, the literature regarding EAS interventions for veterans is sparce, and few rigorous studies have been published. Thus, this review incorporated some studies from the general AAI literature. However, there are differences between EASs and other AAIs, including the setting [113]. Thus, the literature from other AAIs may not be fully generalizable to EASs. Nonetheless, a primary goal of this narrative review was to identify gaps in our understanding and propose research areas and priorities to move the field forward. Those goals have been accomplished.

## 5. Conclusions

Studies from the general AAI literature, as well as studies of EAS interventions, are promising regarding possible enhancement of treatment engagement and/or therapeutic alliance, as well as being associated with several transdiagnostic benefits and symptom reduction. Future rigorous studies are warranted to evaluate both outcomes and underlying mechanisms of action.

## Figures and Tables

**Table 1 ijerph-20-06377-t001:** Investigations of EAS interventions for veterans with trauma histories.

Authors	Intervention	Study Design	Sample Size	Participant Diagnoses	Hypothesized MOA and/or Theoretical Basis for Intervention Benefits for Humans	Outcomes	Other
Arnon et al. [46]	EAT-PTSD, a manualized, eight 90 min weekly sessions, group psychoeducation and horsemanship skills training (groundwork only)	Pre- to post- intervention NR/NC	8	PTSD and psychiatric comorbidities	None given	PTSD SX depressive SX No adverse outcomes	Lack of persistent benefit at 3 months
Burton et al. [47]	6-week, metaphor-based group (groundwork only)	Two-arm parallel group NR, PTSD TAU = control group	20 (10 per group)	PTSD	Metaphor	↓ PTSD SX ↑resilience No change in salivatory cortisol	No significant difference between intervention and control groups on PTSD or resilience scores
Duncan et al. [48]	*Can Praxis*, a 7-day, group EAL for couples	Post-NR/NC	31 Canadian veterans and 27 partners	PTSD (for veteran participants)	Principles of effective communication and conflict resolution and healing through mindfulness, cognitive reframes, and somatic approaches	↓ PTSD SX	Used measures that were under development and not fully validated
Ferruolo [49]	One- or two-day group intervention consisting of psychoeducation, experiential equine activities, and group processing	Pre- to post-NR/NC Qualitative	8	Not given	Not given	Locally developed survey revealed themes of learning about self, spiritual connection, trust, and respect.	
Fisher et al. [50]	*EAT-PTSD*, manualized, eight 90 min weekly sessions, group psychoeducation and horsemanship skills training (groundwork only)	Pre- to post-NR/NC	63	PTSD	Not given	↓ PTSD SX ↓ depressive SX	Safe and feasible, benefits persisted 3 months post-intervention
Gehrke et al. [110]	Eight weekly sessions of 3 h each (mounted and groundwork)	Pre- to post-NR/NC	17	PTSD	Not given	↑ Effect ↑ HRV	
Gehrke et al. [109]	Eight weekly sessions of 3 h each (mounted and groundwork)	Pre- to post-NR/NC mixed methods	9	PTSD		↑ Effect Theme clusters were positive impact, connection with the horse, being present, horse mirroring, translating, and power dynamic	
Hoopes et al. [51]	One-time recreational trail ride, approximately 2 h duration	Pre- to post-NR/NC	18	Addictive disorders and PTSD	Biophilia	Positive effect Negative effect Anxiety Craving No adverse outcomes	78% had PTSD, no change in resilience
Johnson et al. [52]	6-week therapeutic horseback riding, conducted once per week (ground and mounted work)	C/R, WL = control group	29	PTSD	Enhanced self-efficacy based upon social cognitive theory	↓ PTSD SX No adverse outcomes	No change coping, self-efficacy, emotional regulation, or perceived loneliness
Lanning et al. [53]	*Rainier Therapeutic Riding’s Riding Through Recovery*, 8-weeks, 90 min sessions (ground and mounted work)	Repeated measures, comparison group = TAU NR	89	PTSD	Natural horsemanship and developing a mutually respectful horse–human relationship	↓ PTSD SX ↓ depressive SX ↓ functional disability	Benefits sustained 2 months after intervention
Lanning et al. [54]	8-weeks, 90 min sessions (ground and mounted work)	Multi-method, repeated measures, NR/NC	51	PTSD	Experiential learning	↓ PTSD SX ↑functioning	Benefits sustained 2 months after intervention
Malinowski et al. [55]	Five one-hour sessions over five days (ground and mounted work)	Pre- to post-NR/NC	7 veterans and 9 equines	PTSD	None given	Humans: ↓ PTSD SX ↓ PD ↓ Blood pressure on one day Equines: No change in cortisol ↓ Heart rate No change in HRV No change in oxytocin	Horse and human physiological, and human psychological, data were collected
Marchand et al. [56]	One 4 h session of EAL/PIH for veterans enrolled in VA residential substance abuse treatment (groundwork only)	Pre- to post-NR/NC	33	Addictive disorders and PTSD	Developing a mutually respectful horse–human relationship	↑ Positive effect ↓ Negative effect ↓ Anxiety ↓ Craving No adverse effects	52% of participants had PTSD, 75% had a history of increased suicidal risk, previous high risk of suicide predicted response
Marchand et al. [57]	Two sessions of horsemanship skills training and two trail rides (ground and mounted work)	Pre- to post-NR/NC	18	PTSD and many had psychiatric comorbidity	Horsemanship skills training, nature exposure	↑ Positive effect ↓ Negative effect ↓ Depressive SX ↑ Psych flexibility ↓ PTSD SX Enjoyed activity No adverse effects	Improved psych flexibility and depressive and PTSD SX persisted for 30 days post-intervention, no changes on quality-of-life measure
Marchand et al. [58]	Whispers with Horses, a six-session manualized intervention providing mindfulness and self-compassion training in the context of a developing horse–human relationship offered as individual and group therapy (groundwork only)	Pre- to post-NR/NC	33	All had trauma histories, 73% had PTSD, many had additional psychiatric comorbidity	Enhanced mindfulness and self-compassion skills and horse–human bonding	↑ Positive effect ↓ Negative effect ↓ Depressive SX ↑ Psych flexibility	Pre- to post-session data did not reveal changes for all sessions, p re- to post-intervention data revealed decreased depressive SX and increased psych flexibility but no change in PTSD SX
Meyer and Sartori [33]	Very limited description	Qualitative study	5	PTSD	Attachment theory	Themes were positive changes in thoughts and behaviors, beliefs about horses’ cognition and emotions, EAS-induced emotions and emotional regulation, and interpersonal and interspecies relationships	
Monroe et al. [59]	Eight-week group intervention with 3 h sessions (ground and mounted work) utilized components of CBT	Pre- to post-NR/NC	48	Self-identified as having PTSD, 76% met diagnostic criteria	None given	↓ PTSD SX ↓ Anxiety ↓ Depressive SX ↑ Quality of life	
Romaniuk et al. [60]	Separate individual and couples group therapy. Based on relational Gestalt therapy, mindfulness, grounding techniques, and natural horsemanship. Couple group for veterans and partners	Within-subjects longitudinal study NR/NC	25 veterans 22 Couples	Not, given but many experienced PTSD symptoms	None given	↓ PTSD SX ↓ Anxiety ↓ Stress ↓ Depressive SX ↑ Quality of life ↑ happiness	Veterans of Australian Defense Force, symptom improvement, except anxiety, maintained at three months post-intervention only for the couple’s cohort
Rosing et al. [108]	Group, 3 h/week for 6 months (ground and mounted work), included Eagala model	Pre- to post-NR/NC qualitative	13	PTSD	None given	Themes were the ability to relax, forming relationships and transformation and hope	Israeli military and police veterans with PTSD
Shelef et al. [64]	Group, 3 h/week for 6 months (ground and mounted work), included Eagala model	Open case series NR/NC	23	PTSD	Equine interaction and group processing	↓ PTSD SX ↑ Functioning No adverse outcomes	
Steele et al. [61]	Trauma and Resiliency Resources, Inc.’s Warrior Camp is a 7-day intensive intervention program, including EMDR, PIH, yoga, and narrative writing (groundwork only)	Pre- to post-NR/NC	85	Not given but all participants were military veterans, and most had been deployed to a combat zone	Equine element may have facilitated a sense of safety and enhanced development of trust, self-esteem, and increased self-efficacy	↓ PTSD SX ↓ Depressive SX ↓ Dissociation ↓ Moral injury ↑ Attachment	PIH was provided using the Eagala model
Sylvia et al. [62]	2-day retreat, veterans and family members participated in three 2 h sessions of EAS (ground and mounted work)	Post-NR/NC	62 veterans 44 family	PTSD was primary diagnosis for 53 participants; other diagnoses were TBI, AUD, and depression	None given	Participants enjoyed the program	Veterans were participating in a two-week intensive treatment program; qualitative data were also reported
Wharton et al. [26]	Equine-facilitated cognitive processing therapy (12 sessions)	Pre- to post-NR/NC	27	PTSD	Cognitive processing therapy	↓ PTSD SX ↓ Trauma-related guilt	
Zhu et al. [63]	EAT-PTSD, a manualized, eight 90 min weekly sessions, group psychoeducation and horsemanship skills training (groundwork only)	Pre- to post-NR/NC functional and structural neuroimaging and DTI	19	PTSD		↓ PTSD SX ↓ Depressive SX ↑ Caudate FC ↓ Gray matter density of thalamus and caudate. The increase in caudate FC was associated with clinical improvement	A longitudinal brain imaging study, including structural imaging, fMRI, and DTI. Subjects were a subset of the subjects reported in the Fisher et al. article above

C/R = controlled/randomized; WL = wait list; R = randomized; PTSD = posttraumatic stress disorder; SX = symptoms; PD = Psychological distress; NR = non-randomized; NR/NC = non-randomized/non-controlled; MOA = mechanism of action; EAL = equine-assisted learning; HRV = heart rate variability; EAL = equine-assisted learning; PIH = psychotherapy incorporating horses, psych = psychological; EMDR = eye movement desensitization and reprocessing therapy; TBI = traumatic brain injury, AUD = alcohol use disorder, fMRI = functional magnetic resonance imaging, DTI = diffusion tensor imaging, FC = functional connectivity, CBT = Cognitive Behavioral Therapy, HRV = heart rate variability, ↑ = increase, ↓ = decrease.

**Table 2 ijerph-20-06377-t002:** Key research areas and priorities.

Area	Current State	Research Priorities
Manualized EAS interventions	Currently a few manualized interventions exist, but no comparison studies have been conducted.	Direct comparisons of interventions in randomized controlled trials including multi-site studies.
Dose–response relationships and frequency of treatment	The optimal dosage and frequency of EAS interventions is unknown.	Compare outcomes of standardized interventions across multiple dosing and frequency options.
Horse–human interaction	Limited understanding of mechanisms underlying how horse–human interactions contribute to therapeutic outcomes. Relative benefits of EAS interventions with mental-health-treatment components versus those without is unknown. Relative benefits of EAS interventions with mental-health-treatment components that use metaphor versus those that do not use metaphor is unknown. Relative benefits of EAS interventions with mounted activities versus those without is unknown.	Mechanism of action studies measuring human attachment, psychological flexibility, mindfulness, human–animal attachment, and sense of control, autonomy, and assertiveness. Parse the potential mechanism of biophilia. Mechanism of action studies measuring horse and human HRV, heartrate synchronization, oxytocin, and cortisol, as well as human brain activation. Compare outcomes of standardized interventions with and without a mental-health-treatment component. Compare outcomes of standardized interventions with a mental-health-treatment component and with and without the use of metaphor. Compare outcomes of standardized interventions with and without mounted activities.
Treatment engagement and therapeutic alliance with EASs and conventional interventions	Most evidence from the general AAI literature. Some evidence that veterans enjoy EASs and attrition may be less that with conventional interventions.	Direct comparisons of enrollment and attrition in EASs versus conventional treatments. Utilization of treatment engagement and therapeutic alliance scales. Qualitative studies of veteran EAS experience. Determine if physiologic or psychological measures correlate with treatment engagement and/or therapeutic alliance in EASs. Determine if human–horse attachment is correlated with treatment engagement and/or therapeutic alliance.
Transdiagnostic benefits and symptom reduction	Many studies suggest benefit, but rigorous studies are generally lacking. Limited evidence of long-term benefit.	Large randomized controlled trails. Determine if human–horse attachment correlates with treatment response. Determine if physiologic measures correlate with treatment response.
Potential adverse outcomes for human participants	Horse-related injuries, short-term emotional discomfort, and re-traumatization are risks of EASs. A few EAS studies of veterans have reported no adverse outcomes.	EAS interventions should have a response to adverse events plan. Studies should assess for, and report on, adverse outcomes.
Potential adverse outcomes for equine partners	Several studies suggest that EASs are not stressful for horses; however, more research is needed to confirm.	Horse and human physiologic measures, which are useful to disambiguate mechanisms of action, can also be used to evaluate equine stress in EASs. These include HRV, cortisol, and equine behavior scales.

## Data Availability

New data were not generated for this study.

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
