# Peer review of "Potential Mechanisms of Action and Outcomes of Equine-Assisted Services for Veterans with a History of Trauma: A Narrative Review of the Literature"

_ijerph, 2023, doi:10.3390/ijerph20146377_

Round 1
Reviewer 1 Report
General comment: The manuscript “Potential mechanisms of action and outcomes of equine-assisted services for veterans with a history of trauma: A narrative review of the literature” is an ambitious paper having the aim to review the literature of animal-assisted interventions and equine-assisted services (EAS), with the declared goal of addressing the gaps concerning the limited evidence of their benefit as complementary interventions for military veterans who have experienced trauma; and, in the same time, with the aim of providing guidance for future investigations in the field.
The review, although limited to the application to military trauma, is interesting for the practitioners in the field of EAS applied to psychiatric diseases, largely reporting the current results, and critically discussing the gaps of previous investigations, as well as the challenges and future directions of equine-assisted activities and therapies for veterans with post traumatic stress disorder in the light of results actually achieved. The narrative review targeted the existing literature on the mechanisms of action and associated outcomes of equine-assisted services for veterans with trauma histories, also taking into account the key findings from the literature concerning the equine physiological and behavioral characteristics, with special emphasis concerning those that contribute to the dimensions of horse-human relationships, such as horse sensory cues, behavioral strategies and horse cognition and emotion. The author critically discusses the most important features of EAS programs, such as horse-human relationships, attachment, and bonding, that are usually considered to contribute to outcomes in studies of veterans, and their role in treatment engagement, and therapeutic alliance. Evidences suggesting that EAS interventions are associated with both trans-diagnostic benefits and reduction of symptoms associated with specific psychiatric disorders are reported and commented. The therapeutic responses experienced by humans from horse-human emotional mirroring and heart rate synchronization are also discussed, as well as the role in the therapeutic benefits of EAS of enhanced psychological flexibility, nature exposure, and mindfulness. Physiological measures and outcomes that could provide insights into the mechanisms of EAS interventions have been finally mentioned; and measures include cortisol, oxytocin, heart rate variability (HRV), and brain activation. The paper provides recent bibliographic data and an in-depth discussion of results.
Title: It is correct.
Abstract: It is suitable. It clearly identifies the interest for this study and its possible relevance. It recaps the information contained in the main text.
Introduction: The Introduction provides adequate background and clearly defines the aim of the review. This section includes recent and specific literature references, although limited to the declared aim of the review. The presentation of bibliographic data follows a logical line; it is clear but too extensive. The comments reported are pertinent to the data achieved. The authors critically examine the data in the light of the state of science highlighted in the introduction.
Materials and Methods: This section is clear, correctly and extensively presenting the methodology adopted for the literature review search strategy. The objective is mainly a clinical one and the aim is of great value. Nevertheless, as this is the Author’s aim, the critics relative to the lack of rigorous research in previous investigations are conceited, being the majority of them conducted in the early stage of scientific development of the argument and involving different scientific and specific aspects of the EAS interventions and outcomes. The same consideration is concerning the suggestions of research priorities that really could only be achieved by a large cooperation in the field of scientists with various and large competencies and competent technical staff. On the contrary, a complete agreement regarding the need of continuous studies of physiologic measures to fully disambiguate mechanisms underlying the benefits of horse-human interactions can be provided. Moreover, further information on markers of stress levels in equines and humans, and the effect of various EAS on equines could add value to critical evaluation of data achieved, so to improve further speculative planning of research. A further survey of literature concerning EAS effects could suggest that, in order to widen the perspectives of scientific knowledge, potential measures could include not only both horse and human HRV, cortisol, and oxytocin as well as human blood pressure, skin conductance and brain activation, but also other physiological and hormonal variables. Moreover, it is not clear whether there is a need of remark some phrases in a different type of character and size; or whether, perhaps, there is a carelessness in typing the test. If so, please, do correct!
Key Findings from the Literature: The literature data achieved by the author are logically presented and accompanied by clear tables. The review is certainly interesting, but it is sometime boring. They are extensively described and substantially commented in the light of the aim of the study. The author critically examines the results of data achieved in the light of the state of science highlighted in the introduction and the comments reported are pertinent and follow a logical line. However, many paragraphs could be shortened (e.g., equine characteristics), because it is evident the need to show a complete knowledge of all the aspects of the argument and the content could be overviewed and sections reworked to provide a greater level of synthesis. Please, consider it would be also extremely helpful to have a clear explanation of EAS interventions for veterans with trauma histories reported in the cited interventions (physical, occupational, psychological, speech-language pathology and counselling) utilizing horses as just one of the tools to achieve therapeutic goals, as differentiated from educational activities using horses and adaptive riding. The terminology as described in the paper of Wood et al. (Wood W, Alm K, Benjamin J, Thomas L, Anderson D, Pohl L, Kane M. 2021. Optimal Terminology for Services in the United States That Incorporate Horses to Benefit People: A Consensus Document. THE JOURNAL OF ALTERNATIVE AND COMPLEMENTARY MEDICINE Volume 27, Number 1, 2021, pp. 88-95) should be helpful. Please, overview the current content and rework sections to provide a greater level of synthesis. This will increase the value of this review for readers and for the equitherapy discipline itself. While much useful information has been incorporated in this manuscript, some paragraphs still read as a summary of one paper after another with little synthesizing of the information to come up with new ideas. As it becomes clear from the long list of topics treated, the review tries to take into consideration in a unitary perspective many aspects involving different scientific fields.
Discussion: The discussion of data is balanced and the author’s comments are pertinent to the aim of the review. Potential benefits and potential adverse effects of EAS for humans and equine partners are clearly discussed. Discussion is extensive and follows a logical line according to the aim of the review. Nevertheless, the study presents some limitations. The critics relative to the lack of rigorous research in previous investigations are conceited, being the majority of them conducted in the early stage of scientific development of the argument and involving different scientific and specific aspects of the EAS interventions and outcomes. Limitations of the review are considered.
Conclusions: The conclusions are drawn from the data achieved and are related to the aim of the study. Further information on markers of stress levels in humans and equines and the effect of various EAS could add value to critical evaluation of results obtained. The paper offers the perspective for further study.
References: They are appropriate and present a good up-to-date of items on the argument, although limited to the aim of the review.
Decision: The current manuscript is acceptable for publication after minor revision. The English used in the paper is good and some minor mistakes could be reviewed by the Authors.
Reviewer 2 Report
Thank you for the opportunity to review this paper, which discusses an interesting alternative treatment modality with promise for addressing the ongoing difficulties associated with successfully treating PTSD for military vets. My main feedback concerns the readability and organization of the paper. I believe that the language could be greatly edited and cut down on to more succinctly summarize the state of the literature and future directions.
Introduction
Random font changes throughout.
Bottom of page 2- it is worth mentioning what the medications are.
“VHA was mandated to set aside funds for EAS from its Adaptative Sports Grant program.” Would be helpful to include when this mandate started?
Top of page 4 lines 153-159- a lot of the transitional material in the paper feels repetitive and could be cut down to decrease length and help with overall readability.
Perhaps state why a narrative review and not a systematic review was conducted? “Non-standardized inclusion and exclusion criteria were used” is quite vague.
“Horses must constantly scan the environment for predators and run first, and thus be alive, to ask questions later.” “their visual system is different than ours” Some of the language throughout the manuscript is quite colloquial and could be cut down to shorten the paper.
Is “horsenality” a word? Proof reading for small typos is needed throughout.
“several studies report interventions that are manualized” Make sure the rest of the paper covers and specifies exactly what these manualized treatments are (EAT-PTSD?)
3.2.3. “One way that EAS interventions could benefit veterans with a trauma history would be facilitate engagement in conventional psychotherapy and/or psychopharmacological interventions. In addition to facilitation of engagement in conventional interventions” Are there any studies that have directly tested whether participation in EAS actually facilitates later participation in conventional psychotherapy or psychopharm? If so, should include them, if not should clearly state that they do not yet exist.
3.2.6. I don’t think PIH was previously defined?
3.2.7. If possible I would suggest using words instead of acronyms as this paragraph quickly becomes confusing to read.
p.17 recommend editing the first sentence in line 226 that suggests that biophilia is defined as “nature exposure”
4.2 Manualized EAS interventions for veterans- it seems to me that a (if not the) major contribution of this review would be to summarize the current state of EAS interventions, including a clear delineation and description of current manualized EAS interventions. But, how its written- they are mentioned in the tables but not elaborated on in the text. I would like to see a paragraph that sums up the EAS interventions currently being used, with citations for further information, as well as limitations and what’s needed next. I would imagine that most readers of this paper would be totally unfamiliar with manualized EAS interventions and should walk away from reading this paper with a clear grasp on the current state of the literature. For example, Table 1 mentions equine-facilitated CPT- I think hearing more about this would be of great interest to myself and other readers. To this end, the discussion feels somewhat repetitive with the literature reviewed above, perhaps this could be cut down to talk more about current interventions and future directions needed for the field.
The tables are very useful.
Minor typos and font changes throughout.
Round 2
Reviewer 2 Report
I appreciate the author's responsiveness to reviewer comments. I have no further concerns and recommend the manuscript for publication.